# Mechanistic insights into histone recognition and H3K14 acetylation by the NuA3 histone acetyltransferase complex

Wenping Shi[1,2,5], Lixia Zhao[1,5], Yiru Wang[1,2,5], Yi Zhang[1], Simiao Liu[3], Yannan Wang[1], Roger D. Kornberg [1,4] ✉ & Heqiao Zhang[1,2] ✉

The NuA3 histone acetyltransferase complex in budding yeast, composed of six subunits, specifically acetylates lysine 14 on histone H3 (H3K14), thereby regulating various biological processes. Despite its importance, the structural basis and mechanism underlying histone tail recognition and substrate specificity of the NuA3 complex have remained elusive. Here we report cryo-electron microscopy structures of the NuA3 complex in its apo form, bound to acetyl-coenzyme A (acetyl-CoA), and in a complex with both the histone H3 tail and acetyl-CoA. Our structure shows that the histone tail-binding cleft of NuA3 is formed cooperatively by two subunits, the catalytic subunit Sas3 and the non-catalytic subunit Nto1. A hydrophobic part of the cleft engages the region preceding H3K14 (residues 9-12), while a network of polar interactions between the cleft and the backbone of H3 residues 12-15, particularly involving Gly13, contributes to substrate specificity.

Chromatin organizes the eukaryotic genome by compacting DNA and regulating its accessibility. Its fundamental unit, the nucleosome, consists of ~147 base pairs of DNA wrapped around a histone octamer[1,2]. Flexible N-terminal tails of the histones are subject to a variety of post-translational modifications (PTMs), including methylation, acetylation, phosphorylation, and ubiquitination[3–5]. These modifications are essential for the regulation of gene expression and have wide-ranging effects on cellular processes, including development, genome maintenance, and overall cell physiology[5]. Histone acetylation, one of the most extensively studied modifications, is catalyzed by histone acetyltransferases (HATs) or histone acetyltransferase complexes, which reduce the positive charge of the histone tails and contribute to the activation of gene transcription[6–9]. Yeast histone acetyltransferase complexes include SAGA[10], NuA4[11,12], Hat[13], and NuA3[14–16], while the human complexes include SAGA[17], TIP60[18], BRPF1/2/3-HBO1[19], and JADE1-HBO1 complexes[19,20].

The NuA3 complex in *Saccharomyces cerevisiae*, consisting of Sas3, Nto1, Yng1, Eaf6, Taf14, and Pdp3, is a typical histone acetyltransferase complex in budding yeast and belongs to the MYST (Moz, Ybf2 [Sas3], Sas2, and Tip60) family[15,16,21–23], whose members share a highly conserved MYST domain in their catalytic subunits. The NuA3 complex specifically acetylates lysine 14 on histone H3 (H3K14) in a manner dependent on pre-existing histone marks, including trimethylated H3K4 (H3K4me3) and H3K36 (H3K36me3)[23–26]. Recruitment of NuA3 to chromatin is mediated by the Yng1 and Pdp3 subunits, which recognize distinct histone marks through specialized domains. Yng1 contains an N-terminal region that binds the unmodified H3 tail[26,27] and a C-terminal PHD domain that recognizes H3K4 methylation, which promotes H3K14 acetylation and facilitates transcriptional initiation[28–30]. Pdp3 associates with H3K36me3 through its PWWP domain, thereby mediating NuA3 recruitment to transcription elongation regions[23]. H3K4 methylation and H3K36me3 can recruit NuA3 individually or cooperatively[24–26]. Taf14, a shared subunit of TFIID, TFIIF, and several chromatin-remodeling complexes[31], binds H3K9acyl (acetylated or crotonylated H3K9) via its YEATS domain and collaborates with Yng1 to promote chromatin engagement[32–37]. Its C-terminal

[1]Shanghai Institute for Advanced Immunochemical Studies, ShanghaiTech University, Shanghai, China. [2]School of Life Science and Technology, ShanghaiTech University, Shanghai, China. [3]Institutional Center for Shared Technologies and Facilities, Institute of Genetics and Developmental Biology, Chinese Academy of Sciences, Beijing, China. [4]Department of Structural Biology, Stanford University School of Medicine, Stanford, CA, USA. [5]These authors contributed equally: Wenping Shi, Lixia Zhao, Yiru Wang. ✉e-mail: kornberg@stanford.edu; zhanghq@shanghaitech.edu.cn

ET domain recognizes ET-binding motifs (EBMs) and is essential for complex assembly, transcription, and DNA repair[31,37–40]. Sas3, the catalytic subunit of the NuA3 complex, possesses a MYST domain that directly catalyzes the acetylation of H3K14[22], thereby influencing various cellular processes, including transcriptional activation, cell cycle progression, stress response, and DNA repair[7,25]. Previous structural studies have focused on the catalytic domains of other histone acetyltransferases, such as GCN5[41], Esa1[42,43], MOZ[44], MOF[45], and HBO1[46,47], as well as on the SAGA complex[48–50], the yeast NuA4 complex[51–55], and the human TIP60 complex[56–58]. However, the structural information and mechanism underlying histone tail recognition and substrate specificity of the NuA3 complex has remained largely unknown. Here we report cryo-electron microscopy (cryo-EM) structures of the NuA3 complex that afford insight into histone tail recognition and H3K14 acetylation.

## Results

### Purification, enzymatic activity, and overall structure of the NuA3 complex

The six subunits of the NuA3 HAT complex were cloned into pFastBac1 or pFastBac-dual vectors and co-expressed in insect cells. The resulting complex was subsequently purified to homogeneity by anti-FLAG affinity followed by size-exclusion chromatography (Fig. 1a and Supplementary Fig. 1). In the presence of H3K4 tri-methylation, the purified NuA3 complex exhibited histone acetyltransferase activity specifically at the H3K14 site, as shown by western blot and mass-spectrometry, with nucleosomes and synthetic histone peptides (residues 1-21) as substrates, respectively (Fig. 1b–d).

Cryo-EM analysis was performed with the addition of 0.01% Tween-20 during grid preparation to address preferred orientation issues. Data was collected on a Titan Krios microscope equipped with a K3 detector and processed in cryoSPARC[59], yielding a 3.7 Å reconstruction map (Fig. 1e, Supplementary Fig. 2, Supplementary Fig. 3a-c, and Supplementary Table. 1), which sufficed to build an atomic model of the NuA3 complex (Fig. 1f, Supplementary Fig. 3d, and Supplementary Table 1). All subunits were clearly resolved in the structure except for the Pdp3 subunit, the plant homeodomain (PHD) of Yng1, and the YEATS domain of Taf14 (Fig. 1f, g), which were not observed, due to motion or disorder. Sas3, the catalytic subunit, interacts with Nto1, a non-catalytic subunit, mainly through its N-terminal domain, together forming the core of the NuA3 complex. Two N-terminal α-helices of Yng1, a long C-terminal helix (residues 550-603) of Nto1, and Eaf6 assemble in a helical bundle, positioned on one side of the NuA3 core. On the opposite side, the extra-terminal (ET) domain of Taf14 interacts with the N-terminal domain of Sas3 (Fig. 1f).

### Conserved catalytic features shared by the MYST family members

The MYST domain of Sas3 adopts a canonical histone acetyltransferase fold, characterized by a central β-sheet flanked by α helices. In addition to the MYST domain, the catalytic core, extra N-terminal and C-terminal regions of the Sas3 subunit are also revealed in the structure (Fig. 2a). The overall architecture of the Sas3 MYST domain closely resembles that of yeast Esa1 (PDB entry: 1FY7)[42,43], as well as human HBO1 (PDB entry: 6MAJ)[46,47], MOZ (PDB entry: 9FKR)[44], and MOF (PDB entry: 3TOA)[44,45] (Fig. 2a). The catalytic mechanism of yeast Esa1 is well established, with cysteine and glutamate residues playing essential roles in catalysis[43]. Our structure shows that the catalytic residues, Cys418 and Glu452 in Sas3, are not only conserved at the sequence level (Fig. 2b), but also maintain the same spatial arrangement in the three-dimensional structure (Fig. 2c). Mutation of any catalytic residue abolished the HAT activity of the NuA3 complex (Fig. 2d), attesting to its significance in catalysis.

### The interactions between subunits

The Nto1 subunit is composed of three distinct domains: the N-terminal EPL1-like domain, the central PZP (PHD1-Zn-Kn-PHD2) domain, and the C-terminal helical domain (Fig. 1g). Nto1 extends through a tunnel formed by the catalytic subunit Sas3 and the non-catalytic subunit Yng1, making extensive interactions. The EPL1-like domain is positioned on one side of the tunnel, while the PZP domain lies on the opposite side (Fig. 3a). The cysteine-rich PZP domain has been extensively characterized in previous studies[60–63]. The PZP domain of Nto1 aligns closely with those of BRPF1[60], AF10[61], PHF14[62], and JADE1[63] (Fig. 3b). The Nto1-PZP consists of three subdomains: two PHD subdomains connected by a zinc knuckle (Zn-Kn) subdomain, collectively forming a five-zinc cluster structure (Fig. 3c). Except for one zinc-coordinating motif in PHD1, which consists of four cysteines, the remaining four zinc ions are all coordinated by Cys3His zinc-finger motifs (Fig. 3c). The C-terminal helical subdomain of Nto1 forms a helical bundle with the N-terminal two α-helices of Yng1, Eaf6, and a short α-helix (residues 659-672) of Sas3 (Fig. 3d). In support of previous studies[31,37,40], the ET domain of Taf14 binds to a well-characterized EBM motif present in Sas3 (Fig. 3e, f).

### Recognition and acetylation of histone H3 tail by the NuA3 complex

Using acetyl-CoA as an acetyl group donor, the NuA3 complex selectively acetylates H3K14 when H3K4 or H3K36 is tri-methylated[24–26]. To investigate how the NuA3 complex sequentially binds the co-factor acetyl-CoA and the histone substrate, we first determined the cryo-EM structure of a mutant NuA3 complex (E452Q) bound to acetyl-CoA at 3.1 Å resolution (Fig. 4a, Supplementary Fig. 4, Supplementary Fig. 5, and Supplementary Table 1). The E452Q mutation was introduced to stabilize the complex. The structure shows that the acetyl-CoA, serving as an acetyl group donor, binds to the C-terminal domain of the catalytic subunit Sas3, engaging in extensive polar interactions. Residues Thr421, Arg427, Gly429, Gln432, Ser456, Thr462, and Tyr561 of the Sas3 subunit form hydrogen bonds with acetyl-CoA (Fig. 4b). We then attempted to determine the cryo-EM structure of the NuA3 complex bound to both acetyl-CoA and the histone H3 tail. As noted earlier, previous studies have demonstrated that Taf14 can recognize various histone modifications, including acetylation[32] and crotonylation[34] at H3K9, significantly advancing our understanding of how Taf14 interacts with histone marks. However, the precise role of this interaction in the context of the NuA3 complex remains insufficiently characterized, and a detailed functional understanding is still lacking. In this study, to enable a clear structural analysis, we utilized a synthetic histone H3 peptide (residues 1–21) carrying tri-methylation at H3K4 and a K14M mutation for cryo-EM studies. This synthetic peptide was incubated with the complex and acetyl-CoA, and then analyzed by cryo-EM, resulting in a 3.2 Å reconstruction map (Supplementary Fig. 6, Supplementary Fig. 7, and Supplementary Table 1). Superposition of the NuA3 complex structure reported here with that previously reported for the NuA4 complex[51], which mostly acetylates the H4 histone tail[11,12], revealed substantial conformational differences, particularly in the helical bundle regions (Supplementary Fig. 8a). Whereas the NuA4 complex features a shallow binding cleft formed solely by its catalytic subunit, Esa1 (Supplementary Fig. 8b), our structure of the NuA3 complex reveals a deeper histone H3 tail binding cleft, formed by both the catalytic subunit, Sas3, and the non-catalytic subunit, Nto1 (Fig. 4c, d). The cryo-EM map reveals well-defined electron densities for the histone tail (residues 9-18) and acetyl-CoA. The two catalytic residues, Cys418 and Glu452, are positioned between the 14th residue of the histone tail and the acetyl group of acetyl-CoA (Fig. 4e), suggesting a pre-reaction state.

The interaction between the histone tail and its binding cleft is primarily mediated by hydrophobic interactions and hydrogen bonds

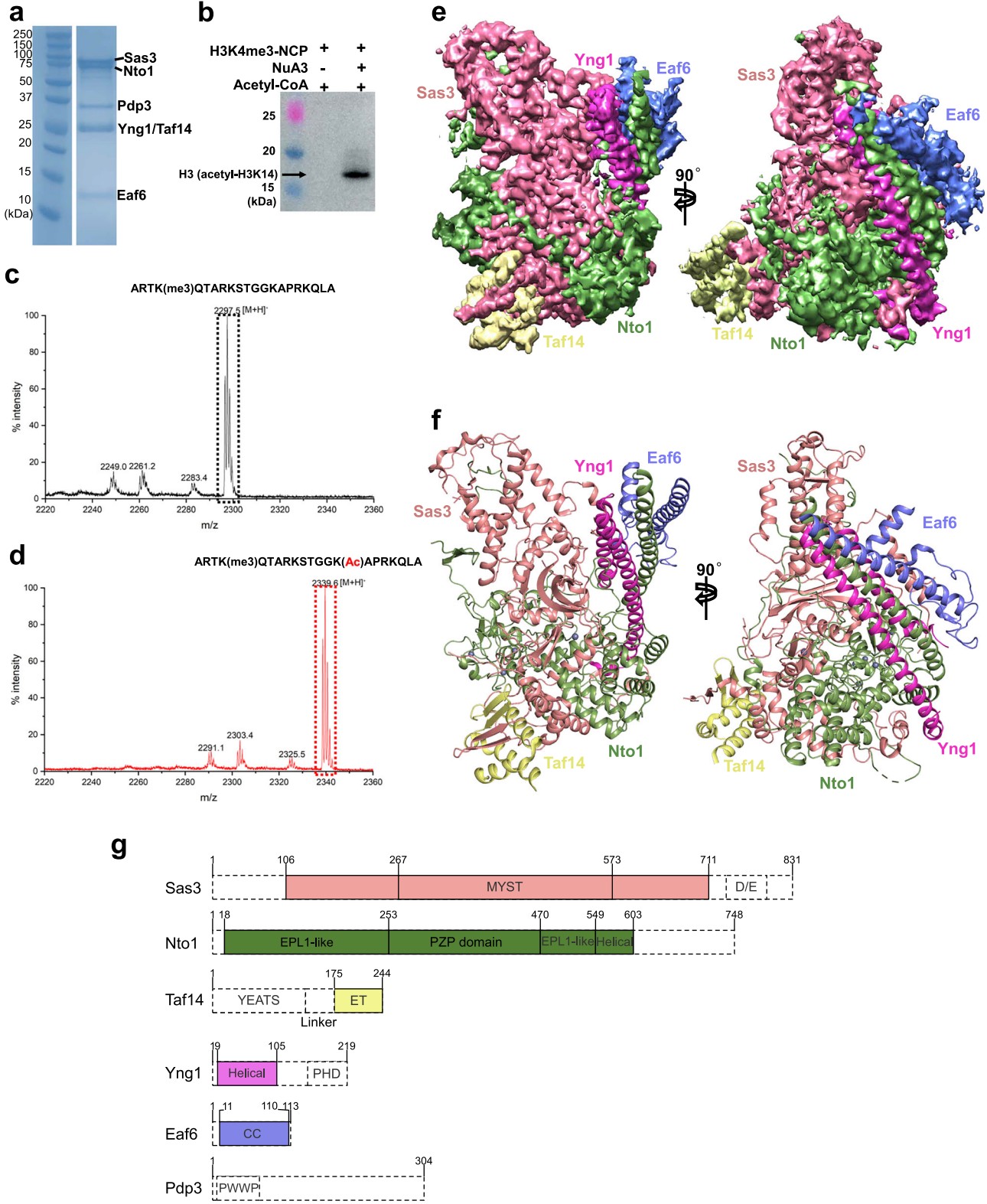

**Fig. 1 | Biochemical and structural characterization of the NuA3 complex in the apo state. a** SDS-PAGE analysis of the purified NuA3 complex. Individual subunits are labeled on the right. Data are representative of three independent experiments. Source data are provided as a Source Data file. **b** HAT activity assay of the NuA3 complex detected by western blot using an anti-H3K14ac antibody. Data are representative of three independent experiments. Source data are provided as a Source Data file. MALDI-TOF analysis of histone H3 (residues 1–21) peptides in the absence (**c**) and presence (**d**) of the NuA3 complex. Unmodified and acetylated peptides are marked with black and red dashed boxes, respectively. Source data are provided as a Source Data file. **e** Segmented cryo-EM maps of the NuA3 complex in the apo state, rotated by 90°, with subunits color-coded. **f** Overall structure of the NuA3 complex, rotated by 90°, with subunits colored as in (**e**). **g** Domain organization of NuA3 subunits. Regions not resolved in the structure are indicated by dashed boxes. MYST(MOZ, Ybf2/Sas3, Sas2, TIP60), MYST histone acetyltransferase domain, D/E, Asp/Glu, EPL1-like, Enhancer of Polycomb-like Protein 1-like, PZP, PHD-Zn knuckle-PHD, ET, Extra-Terminal, PHD, Plant Homeo Domain, CC, Coiled Coil, PWWP, proline-tryptophan-tryptophan-proline.

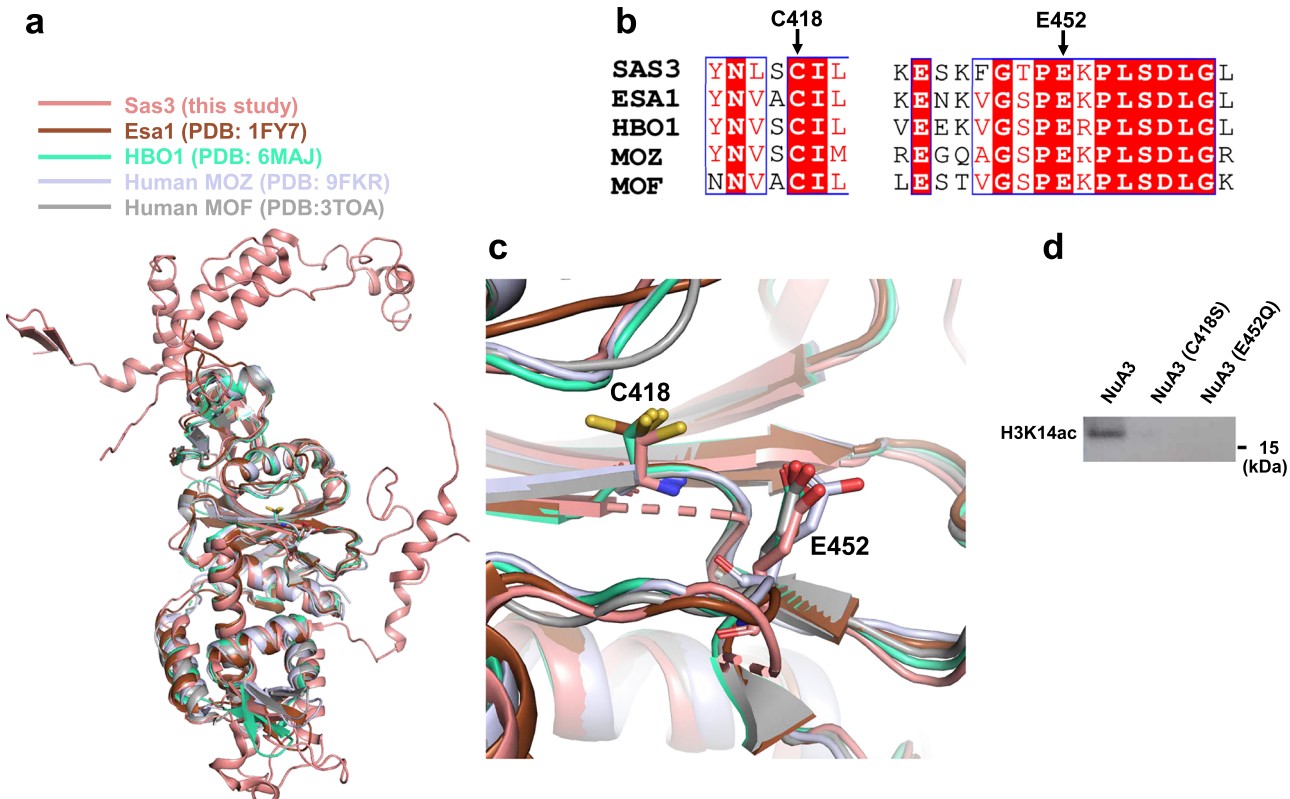

**Fig. 2 | Conserved catalytic features of the NuA3 complex. a** Structural super-position of the NuA3 complex with Esa1, HBO1, MOZ, and MOF. Sas3, Esa1, HBO1, MOZ, and MOF are colored in salmon, chocolate, lime green, light purple, and gray, respectively. **b** Sequence alignment of Sas3 with Esa1, HBO1, MOZ, and MOF. Conserved catalytic residues are marked with black arrows. **c** Superposition of the catalytic residues from NuA3 and other MYST family HATs, shown in stick representation. **d** Western blot analysis of the wild-type and two mutant forms of the NuA3 complex. Data are representative of three independent experiments. Source data are provided as a Source Data file.

(Fig. 5a). A hydrophobic core, composed of Sas3 residues Leu369, Pro375, Phe376, Phe409, Phe580, Ile582, and Met585, together with Nto1 residue Ile298, Phe299, and Tyr357, engages with the N-terminal residues (residues 9-12) of the histone tail (Fig. 5b). A loop of the PHD2 subdomain of Nto1 (residues 349-367), along with the C-terminal domain of Sas3, particularly an α-helix, and the following loop from the C-terminal domain of Sas3 (residues 429-457), which we refer to as the 'CoA-engaging helix' and 'histone-engaging loop', respectively, play essential roles in stabilizing interactions with both acetyl-CoA and the histone tail. Specifically, a hydrogen bond is formed between Asn354 of Nto1 and the backbone of H3-Met14, and two additional hydrogen bonds are formed between Ser366 of Sas3 and the backbone of H3-Ala15. A fourth hydrogen bond is contributed by the backbone of Ser456 within the histone-engaging loop and Arg17 of the histone tail (Fig. 5b). To validate our structural observations, we introduced two point mutations: L369R and N354A. Based on the structure, the L369R substitution is predicted to cause steric clashes with the histone tail, while N354A would disrupt a polar interaction between the Asn354 side chain and the backbone of H3-Met14. In line with these structural predictions, L369R abolished the HAT activity of the NuA3 complex, whereas N354A led to a partial reduction in activity (Fig. 5c).

To validate our structural observations and in vitro enzymatic assays, we introduced two single mutations, E452Q and L369R, which have been shown to abolish the catalytic activity of NuA3 in our assays, into the *SAS3* gene. We then assessed their effects on acetylation levels in *S. cerevisiae* using Western blot analysis. Consistent with our in vitro results, these mutations led to a decrease in acetylation levels compared to wild-type yeast strains (Fig. 5d).

## Conformational changes of the NuA3 complex induced by acetyl-CoA and histone tail binding

Superposition of the NuA3 complex in its apo state, acetyl-CoA-bound state, and histone tail plus acetyl-CoA-bound state reveals conformational changes (Fig. 5e). Upon acetyl-CoA binding, the histone-engaging loop in the NuA3 complex shifts approximately 2.5 Å away from the histone tail binding cleft, and the CoA-engaging helix undergoes a conformational rearrangement (Fig. 5f). By contrast, the conformations of the histone-engaging loop and the CoA-engaging helix remain nearly unchanged upon histone tail binding compared to their conformations in the acetyl-CoA-bound state. These movements likely expand both the histone tail binding cleft, thereby facilitating the subsequent entry of the incoming histone tail. Given that the histone-engaging loop is in proximity to the histone tail, and one of the catalytic residues, Glu452, is located within this loop, its conformational change upon acetyl-CoA binding may reposition the catalytic residue to facilitate catalysis.

## Effect of neighboring residue of H3K14 on the recognition and acetylation of histone tail

Compared to other histone acetyltransferase complexes, such as SAGA and NuA4, which target multiple acetylation sites on histone tails[7], the NuA3 complex, as noted earlier, exclusively acetylates H3K14, suggesting a higher degree of substrate specificity. Previous studies have shown that the H3K4me3, H3K9ac, and H3K36me3 histone marks are specifically recognized by the Yng1-PHD[28,29], Taf14-YEATS[32], and Pdp3-PWWP domains[23], respectively, thereby preventing their interaction with the histone tail binding cleft of the NuA3 complex. We noticed that the NuA3 acetylation site, H3K14, is preceded by a glycine residue

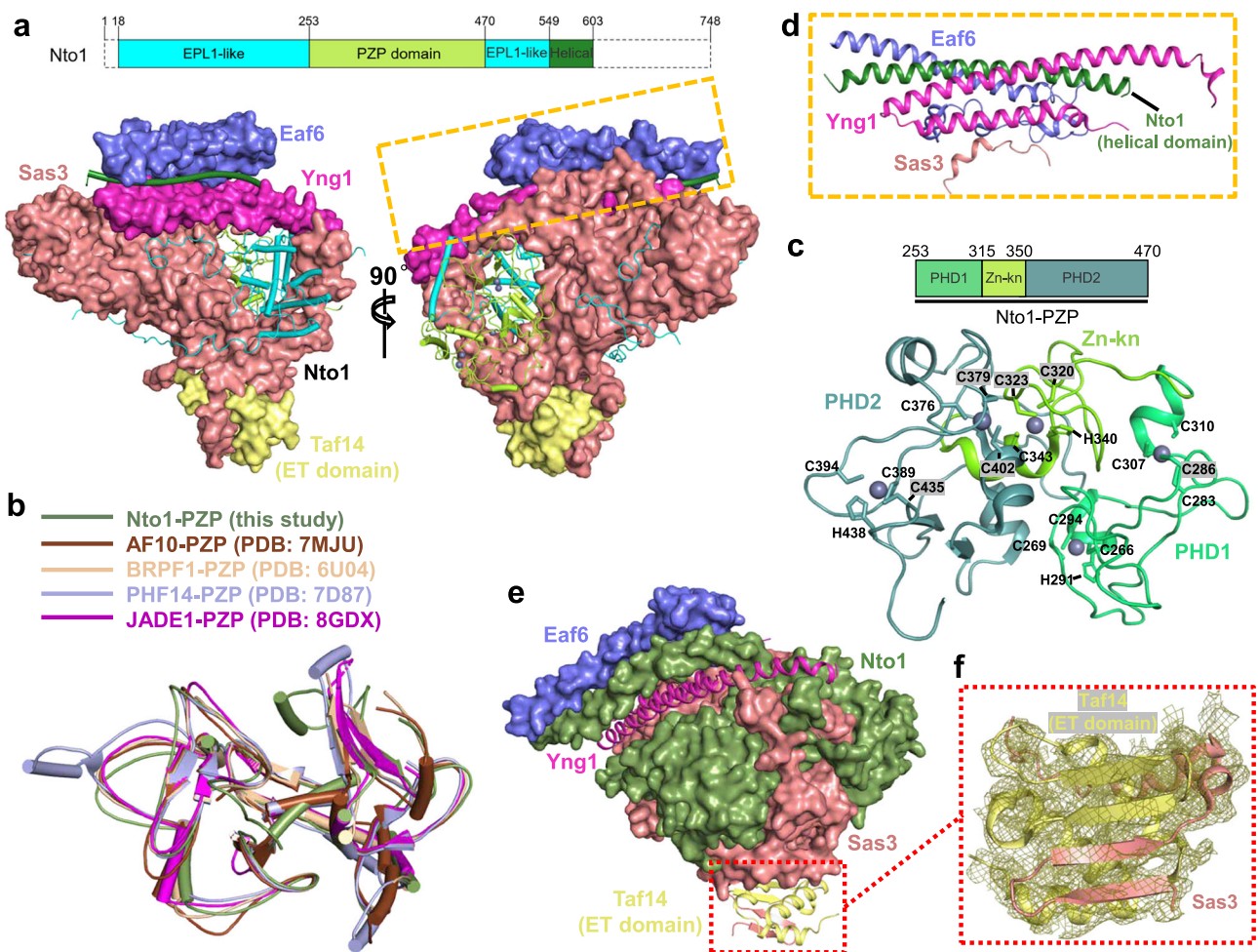

**Fig. 3 | Interaction details within the NuA3 complex. a** Interaction between Nto1 and other subunits. Nto1 is shown in cartoon and the other subunits are shown in surface representation. The top panel shows a schematic diagram of the domain organization of Nto1. **b** Structural superposition of the PZP domain of Nto1 with other PZP domains. **c** Structure of the PZP domain of Nto1, with zinc ions shown as gray spheres. The top panel depicts the subdomain organization of the Nto1-PZP domain. **d** Helical bundle formed by Nto1, Yng1, Eaf6, and Sas3. **e** Interaction between Taf14 and Sas3. The ET-domain of Taf14 and two β-strands of Sas3 are shown in ribbon representation; other subunits are shown as surface. **f** Local electron density of the Taf14-ET in complex with Sas3.

(Gly13), whereas alternative sites not acetylated by NuA3, such as H3K9, H3K18, and H3K27, are preceded by an arginine residue (Fig. 6a). To investigate the role of Gly13 in the specificity of acetylation, we introduced a glycine-to-arginine substitution (G13R) and evaluated its effects with the use of peptide acetylation assay. We expected the G13R mutation would cause steric clashes with the binding pocket, particularly with Leu369 of Sas3 and Tyr357 of Nto1 (Fig. 6b). In line with expectation, our analyses revealed compromised histone acetyltransferase activity of NuA3 toward H3G13R peptides, which were only partially acetylated (Fig. 6c). Given the good shape complementarity between residues H3G12 to H3A15 and the binding pocket formed by Sas3 and Nto1, it is reasonable to propose that, in addition to Gly13, other residues within the histone H3 tail—such as the histone mark H3K9ac (not examined in this study), interactions between H3K4me (residues 1–4) and the Yng1 PHD finger (although not resolved in our structure), as well as N-terminal hydrophobic contacts and hydrogen bonding between H3 and NuA3—may also play a role in the substrate specificity of the NuA3 complex toward its histone substrate.

## Discussion

Our structural studies of NuA3 have shed light on the basis for its exceptional specificity. Many histone-modifying enzymes harbor multiple domains capable of recognizing histone tails. In the NuA3 complex, the Nto1 subunit contains a PZP domain within which reside two PHD subdomains, the Yng1 subunit also harbors a PHD domain, the Taf14 subunit includes a YEATS domain known to bind acetylated histone tails, and the Pdp3 subunit contains a PWWP domain that specifically recognizes the H3K36me3 mark.

In our structure, Sas3 interacts with Taf14, whereas the neighboring residues of the EBM motif in Yng1 are located far from the ET domain. It is possible that conformational changes under certain conditions might allow Yng1 to interact with Taf14, and further studies are needed to test this possibility.

The human MYST family HAT complexes—such as the BRPF1-HBO1 and BRPF1-MOZ/MORF complexes—lack counterparts of the Taf14 and Pdp3 subunits of yeast NuA3. We were unable to resolve the structure of either the BRPF1-HBO1 or BRPF2-HBO1 complexes by cryo-EM. Instead, we employed AlphaFold 3[64], guided by sequence alignment and the cryo-EM structure of the NuA3 core region presented in this study, to predict the core structures of two human homologs of NuA3, the BRPF1-HBO1-ING5-MEAF6 and the BRPF2-HBO1-ING5-MEAF6 complexes. Despite sequence-level divergence, the predicted structures of the BRPF1/2-HBO1-ING5-MEAF6 complexes exhibit an overall architecture similar to that of the NuA3 complex reported in this study (Fig. 7a–e). Structural comparisons revealed similar binding clefts formed between the catalytic subunit HBO1 and the non-catalytic

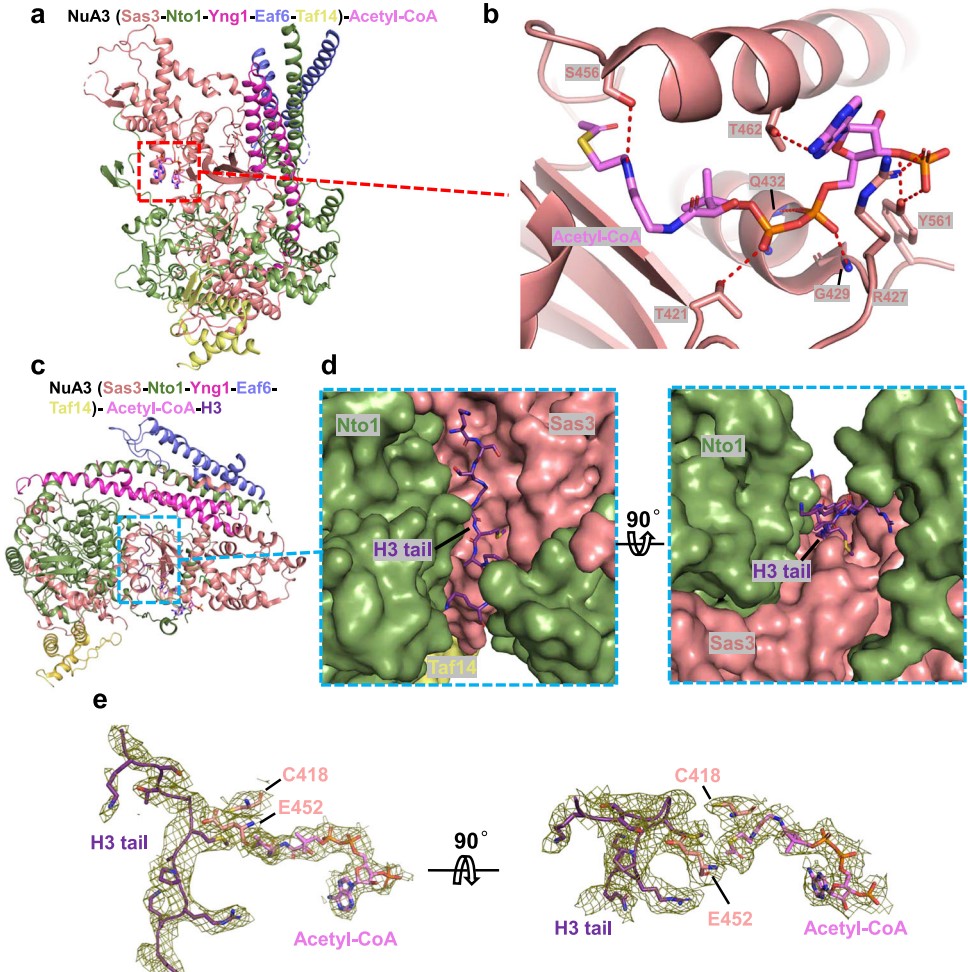

**Fig. 4 | Structural analysis of NuA3 bound to histone H3 tail and acetyl-CoA.**
**a** Overall structure of the NuA3 complex bound to acetyl-CoA. Different subunits are color-coded. **b** Interaction between acetyl-CoA and the NuA3 complex. Interacting residues and acetyl-CoA are shown in stick representation. The local electron density corresponding to acetyl-CoA is displayed in mesh. **c** Overall structure of the NuA3 complex bound to acetyl-CoA and histone tail. Different subunits are color-coded. **d** Binding of the histone H3 tail to NuA3. The histone H3 tail is shown in ribbon representation, and Sas3 and Nto1 are shown as surface, rotated by 90°. **e** Local electron density of the histone tail, acetyl-CoA, and the two catalytic residues of the NuA3 complex, rotated by 90 °.

subunits BRPF1/2, which are likely critical for histone tail recognition and acetylation (Fig. 7f, g). Moreover, a loop containing a Phe-x-Asn (FxN) motif within Nto1 (residues 350–362), designated the 'FxN loop', engages in histone tail interaction, appears to be structurally conserved in the human homologs BRPF1 and BRPF2, as similar loops are present in their predicted structures (Fig. 7h). Sequence alignment further supports this conservation at the amino acid level (Fig. 7i). This structural correlation remains to be experimentally validated in future studies.

In this study, we present cryo-EM structures of the NuA3 histone acetyltransferase complex in its apo form and in complex with the histone tail and acetyl-CoA. Previous studies have indicated that NuA3 exists in two functionally distinct forms: NuA3a, which is recruited to promoter regions of actively transcribed genes through recognition of the H3K4me3 mark by the Yng1 PHD domain to facilitate transcription initiation, and NuA3b, which targets gene bodies via the Pdp3 PWWP domain's interaction with H3K36me3, promoting transcription elongation[23]. Despite considerable efforts, we were unable to determine the structures of NuA3 bound to nucleosomes bearing either H3K4me3 or H3K36me3 modifications. Moreover, the Yng1 PHD domain and Pdp3 subunit are not resolved in our cryo-EM maps. Consequently, this study does not provide structural evidence to differentiate between the NuA3a and NuA3b forms.

## Methods

### Expression and purification of yeast NuA3 complex

The six-subunit *S. cerevisiae* NuA3 complex, including the subunits *SAS3* (UniProt: P34218), *NTO1* (UniProt: Q12311), *PDP3* (UniProt: Q06188), *YNG1* (UniProt: Q08465), and *EAF6* (UniProt: P47128), was reconstituted via recombinant expression. Genes encoding these subunits were PCR-amplified from genomic DNA of the BY4741 yeast strain. A truncated Sas3 construct (residues 105–831), fused to a C-terminal TEV cleavage site and FLAG tag, was co-cloned with *EAF6* into a pFastBac-dual vector using homologous recombination (ClonExpress Ultra One Step Cloning Kit V2, Vazyme). *YNG1*, bearing a C-terminal 3 C cleavage site and 6×His tag, was co-cloned with *NTO1* using the same approach. *PDP3* was inserted into an individual pFastBac-dual vector. The oligonucleotide primers used are listed in Supplementary Table 2. A codon-optimized *TAF14* gene, fused by an N-terminal His-SUMO tag, was synthesized (GenScript Biotech) and cloned into pFastBac1.

Recombinant baculoviruses were generated using the Bac-to-Bac system (Invitrogen). High Five cells (Invitrogen, Cat. No. B85502) cultured in ESF921 medium (expression systems, 96-001-01) (1.0×10⁶ cells/mL) were co-infected with four separate viruses and incubated at 27 °C for 48 hours. Cells were collected by centrifugation (1,000 x g, 20 min, 10 °C), lysed in buffer (20 mM HEPES pH 8.0, 500 mM NaCl,

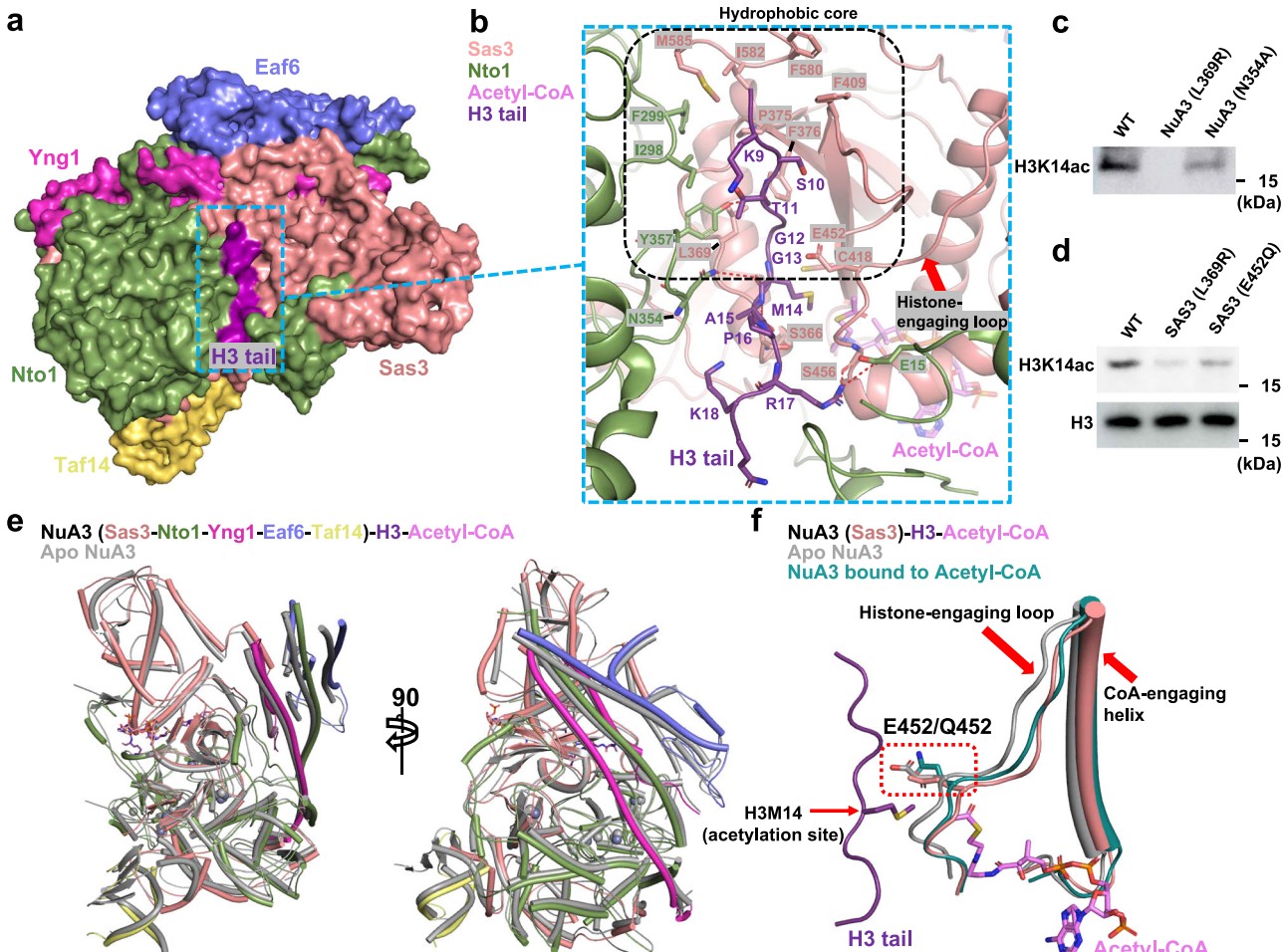

**Fig. 5 | Interaction details between the NuA3 complex, histone H3 tail, and acetyl-CoA. a** Surface representation of the NuA3 complex bound to acetyl-CoA and histone tail. Different subunits are color-coded. **b** Interactions between the histone tail and the NuA3 complex. Interacting residues are shown in stick representation. The hydrophobic core is outlined with a black dashed box, hydrogen bonds are indicated by red dashed lines, and the histone-engaging loop is marked with a red arrow. **c** Western blot analysis of HAT activity in wild-type and mutant NuA3 complexes. Data are representative of three independent experiments. Source data are provided as a Source Data file. **d** Western blot analysis of the acetylation levels in wild-type and *SAS3* mutant *S. cerevisiae* strains, with histone H3 used as the input control. Data are representative of three independent experiments. Source data are provided as a Source Data file. **e** Structural superposition of the NuA3 complex in apo and histone H3-bound states. The apo structure is in gray; subunits in the bound state are color-coded as in Fig. 1. **f** Structural comparison of the NuA3 complex in apo state, acetyl-CoA-bound state, and H3 tail plus acetyl-CoA-bound state. The histone-engaging loop and CoA-engaging helix are indicated by red arrows.

10% glycerol, 2 mM $MgCl_2$, 3 mM DTT, and protease inhibitors), and sonicated. Viscosity was reduced by UltraNuclease (Yeasen) treatment before clearing the lysate by centrifugation (17,420 x g, 1 h, 4 °C).

The supernatant was applied to anti-FLAG G1 resin (GenScript Biotech), washed extensively, and eluted using buffer supplemented with 200 μg/mL FLAG peptide. The His-SUMO tag on Taf14 was removed by Ulp1 protease treatment (overnight at 4 °C). Final purification was performed by size-exclusion chromatography (Superose 6 Increase column, Cytiva) equilibrated in SEC buffer (20 mM HEPES pH 8.0, 150 mM NaCl, 3 mM DTT). The main peak (~15 mL) corresponding to the assembled complex was pooled and concentrated to 0.9 mg/mL for cryo-EM, or 1–2 mg/mL for storage at −80 °C.

Sas3 (L369R, C418S, E452Q) and Nto1 (N354A) point mutants were generated via overlap extension PCR and cloned into pre-validated pFastBac-dual constructs. Expression and purification followed the same protocol as the wild-type complex.

**Cryo-EM grid preparation**
For the apo-state NuA3 structure, 0.9 mg/mL NuA3 complex supplemented with 0.01% Tween-20 was vitrified using a Vitrobot Mark IV (Thermo Fisher Scientific) at 8 °C and 100% humidity. Gold 300-mesh

R1.2/1.3 Quantifoil grids were glow-discharged (30 s, $H_2/O_2$, Gatan Solarus 950). 3 μL of sample was applied, blotted for 2.5 s (blot force −1), and plunge-frozen in liquid ethane.

For the acetyl-CoA-bound complex, 1 mg/mL NuA3 was incubated with 0.5 mM acetyl-CoA in the presence of 0.01% (v/v) Tween-20 on ice for 1 h. For the substrate-bound state, 1 mg/mL NuA3 was incubated with 0.5 mM acetyl-CoA and 0.5 mM synthetic H3K4me3 peptide (21-mer, H3K14M variant; GenScript Biotech) in the presence of 0.01% (v/v) Tween-20 on ice for 1 h. Grid preparations were performed following the same procedure.

**Cryo-EM data collection and image processing**
The apo-state dataset was acquired using a 300 kV Titan Krios microscope (Thermo Fisher Scientific) equipped with an energy filter and a K3 camera, at 105,000× magnification (0.84 Å/pixel). A total of 7,590 micrographs were recorded with a defocus range of −1.2 to −2.5 μm and a total exposure of 50 e⁻/Å². The acetyl-CoA-bound dataset was collected following the same procedure as described above.

For the peptide- and cofactor-bound complex, imaging was performed on a Titan Krios equipped with a Falcon IV detector at 130,000× magnification (0.96 Å/pixel), yielding 4664 micrographs.

**a**

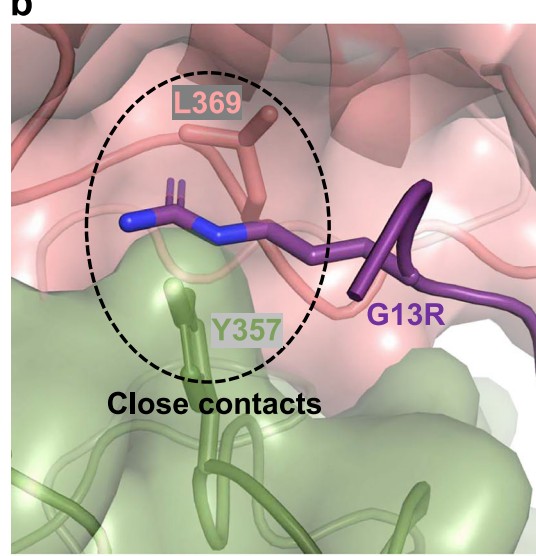

H3K14  GG**K**AP
H3K9   A**R****K**ST
H3K18  P**R****K**QL
H3K23  AS**K**AA
H3K27  A**R****K**SA

**b**

**c**

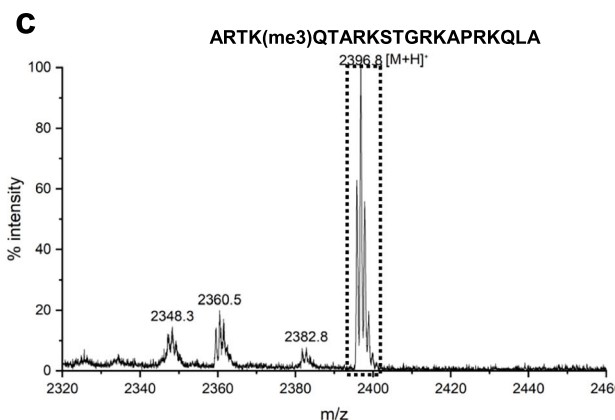

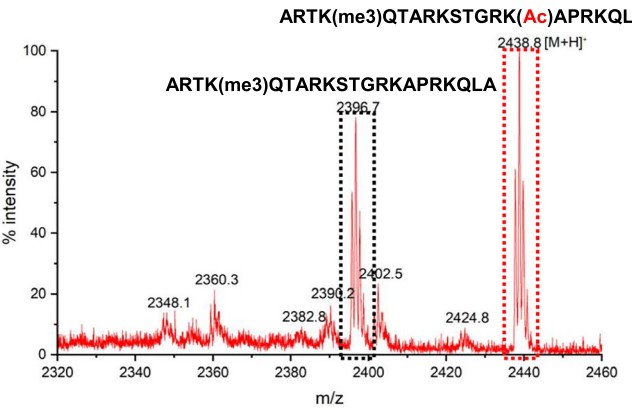

**Partially acetylated**

**Fig. 6 | Impact of the neighboring G13 residue on NuA3 HAT activity. a** Sequence context of different acetylation sites within the histone H3 tail. **b** Predicted steric clashes introduced by the G13R mutation. **c** MALDI-TOF analysis of HAT activity on G13R mutant peptides. Unmodified and acetylated peptides are highlighted with black and red dashed boxes, respectively. Source data are provided as a Source Data file.

Motion correction and CTF estimation were carried out using cryoSPARC's Patch Motion and Patch CTF modules[59]. Particles were initially picked using Blob Picker and classified in 2D. Top 2D classes were used for Template Picker, followed by additional 2D classification and Topaz-based training and picking. Cleaned particle stacks were combined and deduplicated before ab initio modeling and heterogeneous refinement. The best class was subjected to homogeneous and non-uniform refinement, yielding a 3.8 Å map for the apo complex, further improved to 3.7 Å after local refinement (FSC = 0.143). The acetyl-CoA-bound and histone tail plus acetyl-CoA-bound complexes were processed similarly (see Supplementary Figs. 4 and 6), resulting in 3.1 Å and 3.2 Å maps, respectively.

**Model building and figure preparations**

The apo-state atomic model was built into the 3.7 Å map using AlphaFold2[65] predictions as a starting template. Modeling and real-space refinement were performed using Phenix[66], Chimera[67], and COOT[68]. For the acetyl-CoA-bound and acetyl-CoA plus histone tail-bound states, the coordinate file of acetyl-CoA was downloaded from COOT and parameterized via eLBOW in Phenix[66], followed by iterative cycles of real-space refinement and manual adjustment. Figures were prepared with Chimera[67] and PyMOL (www.pymol.org).

**Western blot analysis**

To assess HAT activity, biotinylated H3K4me3 mononucleosomes (Active Motif) were used as substrates. Reactions (15 μL) contained 0.5 μg NuA3 (WT or mutants), 0.5 μg nucleosomes, and 100 μM acetyl-CoA in buffer (25 mM HEPES, pH 8.0, 50 mM NaCl, 3 mM DTT). After 30 min at 30 °C, reactions were quenched with SDS loading buffer, boiled, and resolved on 12% SDS-PAGE. Proteins were transferred to PVDF membranes and probed with anti-H3K14ac (Cell Signaling Technology, Cat. No. 7627 T) followed by HRP-conjugated secondary antibodies. Signals were visualized using ECL substrate and the Amersham Imager 680.

To assess the effects of NuA3 mutants on histone H3K14 acetylation in yeast, site-specific mutations were introduced into the BY4741 strain via homologous recombination. Three DNA fragments (*SAS3*-WT, *SAS3*-L369R, and *SAS3*-E452Q), each containing a *URA3* selectable marker flanked by ~200 bp homologous arms, were synthesized and cloned into pUC19 plasmids (GenScript). Linearized transformation fragments were PCR-amplified from these plasmids and introduced into yeast using the lithium acetate (LiAc) method. The oligonucleotide primers used are listed in Supplementary Table 2. Transformants were selected on SC-URA medium, and successful mutagenesis was confirmed by genomic PCR and sequencing. Verified strains were grown in YPAD to an OD$_{600}$ of ~0.8 for total protein extraction following a standard protocol[69]. Proteins were boiled at 95 °C for 15 min, resolved by 12% SDS−PAGE (15 μL per lane), and transferred to PVDF membranes. Immunoblotting was performed using anti-H3K14ac (Abcam, Cat. No. ab52946) and anti-histone H3 (Selleck, Cat. No. F0057) antibodies, and signals were detected by ECL.

Antibodies used in this study are listed in the Supplementary Table 3.

**MALDI-TOF mass spectrometry**

Acetylation of wild-type and mutant H3 peptides (G12R, G13R, A15Q; GenScript Biotech) was assessed by MALDI-TOF. Reactions

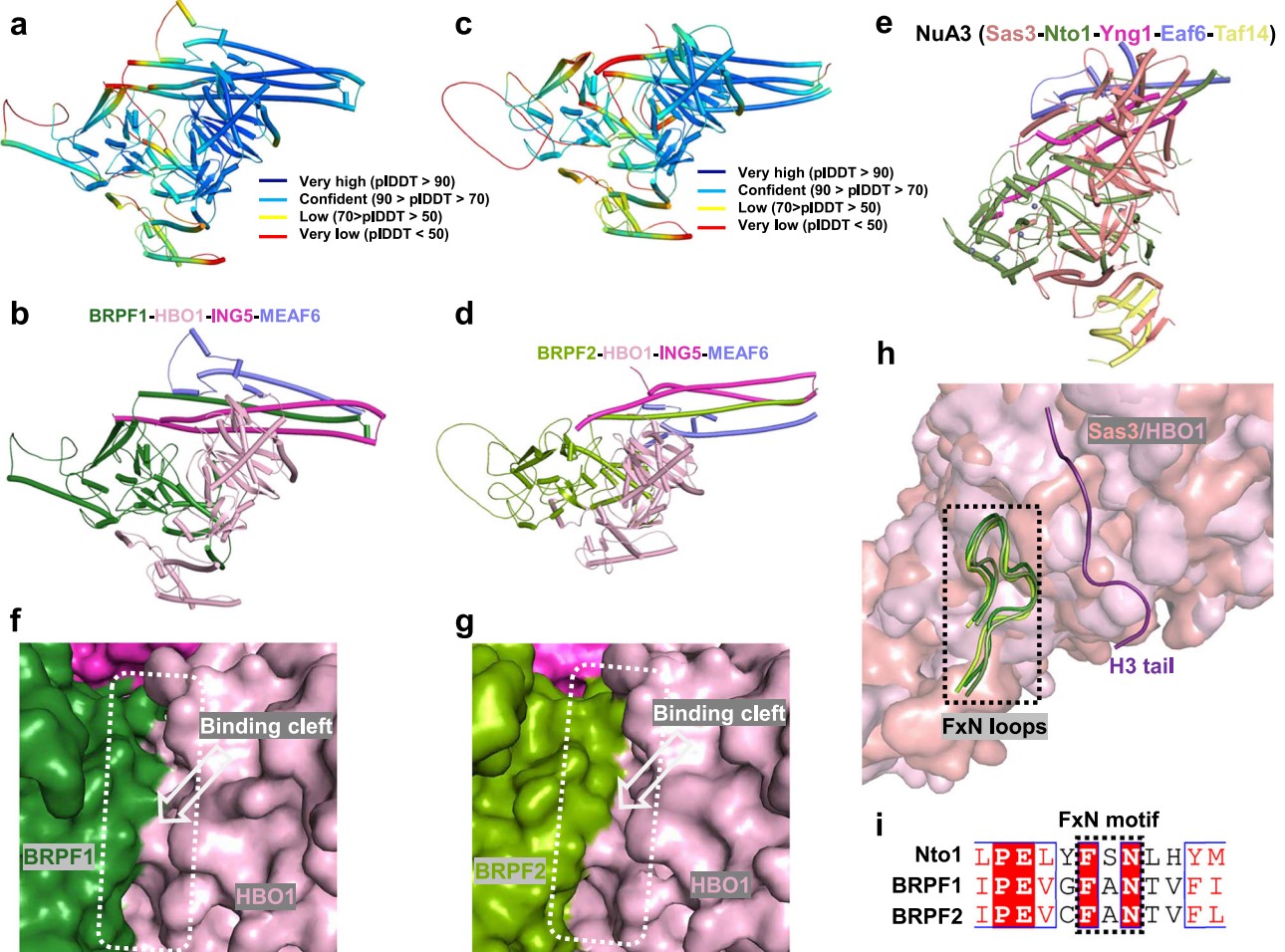

**Fig. 7 | Comparison of binding clefts in MYST family HAT complexes.** AlphaFold3-predicted structure of the BRPF1–HBO1–ING5–MEAF6 complex, shown as pIDDT (**a**) and ribbon (**b**) representations. AlphaFold3-predicted structure of the BRPF2–HBO1–ING5–MEAF6 complex, also shown as pIDDT (**c**) and ribbon (**d**) representations. **e** Cryo-EM structure of the yeast NuA3 complex determined in this study. **f**, **g** Binding clefts formed by BRPF1–HBO1 and BRPF2–HBO1 in the predicted complexes. **h** Structural superposition of the FxN loops from Sas3 and HBO1. Sas3 and HBO1 are shown as surfaces; FxN loops are shown as ribbons. **i** Sequence alignment of FxN loops, with conserved FxN motifs highlighted by a black dashed box.

(20 μL) with 1 mg/mL NuA3, 0.5 mM acetyl-CoA, and 0.5 mM peptide in buffer (20 mM HEPES, pH 8.0, 150 mM NaCl, 3 mM DTT) were incubated at 30 °C for 30 min, then flash-frozen. Samples were spotted with CHCA matrix at a 1:2 sample-to-matrix ratio (0.1% TFA, 50% ACN) and analyzed in positive ion reflection mode on an Autoflex Speed MALDI-TOF/TOF instrument (Bruker).

**Quantification and statistical analysis**

Data obtained from the Autoflex Speed MALDI-TOF/TOF instrument (Bruker) were imported into Microsoft Excel for processing. Peak intensities were normalized relative to the highest-intensity peak (designated as 100%) to establish proportional relationships across the mass spectrum. Subsequently, the normalized data were imported into OriginPro 2025 (OriginLab Corporation) for graphical reconstruction of the mass spectra. Histone acetyltransferase activity assays using nucleosomes as substrates were performed in at least three independent replicates, and representative results are presented.

**Reporting summary**

Further information on research design is available in the Nature Portfolio Reporting Summary linked to this article.

## Data availability

The cryo-EM reconstruction maps for the NuA3 complex in its apo form, bound to acetyl-CoA, and in a complex with both the histone H3 and acetyl-CoA have been deposited in the Electron Microscopy Data Bank (EMDB) under the accession codes EMD-64513, EMD-65145 [https://www.ebi.ac.uk/pdbe/entry/emdb/EMD-64515], and EMD-64517, respectively. Their respective coordinate files have been deposited in the Protein Data Bank (PDB) under the accession numbers 9UUO, 9VKM, and 9UUS, respectively. The AlphaFold3-predicted models of the BRPF1–HBO1 and BRPF2–HBO1 complexes are available in ModelArchive (modelarchive.org) under accession codes ma-c50jd [https://modelarchive.org/doi/10.5452/ma-c50jd] and ma-lqvzp [https://modelarchive.org/doi/10.5452/ma-lqvzp], respectively. The raw mass spectrometry data are included in the Source Data file. Any additional information required to reanalyze the data reported in this paper is available from the corresponding author upon request. Source data are provided with this paper.

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

## Acknowledgements

We thank the Bio-Electron Microscopy Facility of ShanghaiTech University for technical support in cryo-EM data collection. We are also grateful to Ms. Suhong Liu from the Analytical Instrumentation Center of SPST at ShanghaiTech University for assistance with mass spectrometric analyses. We thank Dr. Ying Lei, Ms. Juan Kong, Mr. Pengwei Zhang, and Ms. Lishuang Zhang from the Discovery Technology Platform at SIAIS for their technical support. This research was supported by a grant from ShanghaiTech University to H.Z. and a grant from the Shanxi Key Laboratory of Protein Structure Determination (202104010910006) to H.Z.

## Author contributions

H.Z. conceived and initiated the project; H.Z. and R.D.K. supervised and designed the project; W.S., Yiru Wang, and L.Z. carried out the experiments; W.S., Yiru Wang, L.Z., Y.Z., S.L., Yannan Wang, and H.Z. analyzed the data; Yiru Wang and H.Z. collected and processed the cryo-EM data; H.Z. built and refined all the structural models; H.Z. and R.D.K. wrote the paper with input from all authors.

## Competing interests

The authors declare no competing interests.
