## [Transparent Peer Review file · Nature Communications]

Mechanistic insights into histone recognition and H3K14 acetylation by the NuA3 histone acetyltransferase complex

Corresponding Author: Professor Heqiao Zhang

Version 0:

Reviewer comments:

Reviewer #1

(Remarks to the Author)

In this work, Shi and colleagues report the cryo-EM structure of the yeast NuA3 complex in the apo state and in the presence of acetyl-CoA and histone H3K4me3 peptide (aa 1-21 of H3). Three subunits, Sas3, Nto1 and Eaf6 are fully resolved in the structure, two subunits, Taf14 and Yng1 are partially resolved with the PHD finger and the YEATS domain missing, and no density is observed for the sixth subunit, Pdp3. Despite the missing elements, the structure provides very helpful information regarding some aspects of the complex assembly and how the region of H3 primes at the Sas3 active site for acetylation of H3K14. The major weakness of this work is the incomplete complex structure which may need revision in future. The data on the H3 peptide-acetyl-CoA-Sas3 contact(s) is a solid and important finding here, but it's a relatively small set of results, and I am afraid a considerable number of biological/functional experiments is needed to corroborate the contacts and complete this work. Alternatively, if functional approaches are not feasible, additional structures should be included, which might be more straightforward to this group as the NuA3 complex is already available.

The structural data are of excellent quality, some conclusions are justified though some are unconvincing, still I support this work and believe the manuscript can be effectively reworked and with inclusion of additional data will be a valuable contribution to the epigenetics field.

A few specific comments:

1. Abstract: please remove the phrase "In sharp contrast with previously... complexes". The negative view on others' findings never perceived well, especially if you are somewhat incorrect. And overall, for the entire text, I would suggest writing it in a way to convey how your discovery [taking advantage of the previous knowledge] moves the epi field forward.
2. First sentence of Intro is incorrect and indicates a limited knowledge of the chromatin field.
3. Line 65, mentioning H3K4me3 and H3K36me3- who binds these marks and why these interactions are important for the function of the complex? ref 23-25 are great but quite old papers. Numerous more recent studies have to be discussed here, including those that show that the PHD finger of Yng1 is essential for the NuA3 function(s). The PHD finger is not observed in the current structure, even in the presence of the H3K4me3 peptide, however this cannot be the reason for not discussing Yng1.
4. Similarly to #3, no mentioning of the YEATS domain of Taf14 and the importance of its binding to H3K9acyl for the NuA3 complex.
5. Overall, Intro needs to be substantially expanded with an expert level discussion of up-to-date studies.
6. Lines 94, 98, the word 'extensively' is not needed
7. Fig. 1b- please add labels
8. Fig.1, please add schematics of all six subunits of the NuA3 complex and indicate which parts are observed in the structure.

9. Line 139, please add refs to PZP domain, and because PZP is a well-studied domain, the paragraph lines 138-149 could be condensed into a couple of sentences.

10. Lines 150-161. This paragraph needs to be reworked and shortened to essentially one sentence. Ref 46 discovered that the hxxhx motif (referred to EBM, ET binding motif) is present in both Yng1 and Sas3, and that the ET domain of Taf14 binds EBM-containing peptides derived from both Yng1 and Sas3. Both interactions are thoroughly investigated, and the authors of ref 46 put forward the idea that either Yng1 or Sas3 can be the intra-complex ligands of Taf14. Ref 46 reports the structure of the ET domain with only one of those peptides, Yng1 peptide.

The way how the paragraph on lines 150-161 is written is misleading: omitting Sas3 and focusing on Yng1. Conversely, it should be written 'In support of previous studies (ref 46, please also add Nguyen 2023, BBA, Klein 2022, Nat Comm, and Chen 2020, Nat Comm) the ET domain of Taf14 binds to a well-recognized ... motif, present in Sas3' or something similar.

It will be appropriate in discussion to come back to ref 46 stating that '... from our structure, indeed Sas3 interacts with Taf14 but Yng1 is far away...' or something similar

Also appropriate (and valuable) to discuss possibilities of conformational changes in the complex that would allow Yng1 to interact with Taf14. [why this motif is present in Yng1?] [of note, the same EBM motif is also present in Nto1, unpublished]

11. Line 191, no rationale is given to use H3K4me3 (1-21) peptide. If the rationale was to aid in detecting PHD of Yng1, it needs to be stated. Overall, the use of this peptide is the major weakness of this work. Why not H3K4me3K27acK36me3 or H3K9ac? Or unmodified H3 (to engage with PZP of Nto1)

12. Related to #11, the use of any of these peptides may not provide a true mechanism of the complex function. The nucleosome must be used here, and better PTM containing. In the current structure H3K4me3peptide-acetyl-CoA- bound complex, the only solid unambiguous result is the contacts within the H3(9-18)-Sas3 active site; the rest of the mechanistic data may be completely off in the NCP-complex structure.

13. Line 222, re PZP of Nto1- would be good to superimpose with other PZP domains

14. Line 272, please discuss YEATS domain of Taf14 and its role

15. Lines 275-287, the G13R mutation data to explain NuA3 selectivity are unconvincing and naïve. The RKac motif may play a role for the YEATS domain.

16. Line 305, this phrase is incorrect- PZP domain binds exclusively to unmodified H3; methylation of K4 inhibits this interaction. [I hope that this is a misspelling error rather than the rationale to use H3K4me3 peptide in this work]

17. Lines 309-310, this phrase is again incorrect, and it's unlikely that the PZP of Nto1 lost its ability to bind unmodified H3

18. Overall, I would suggest reworking the manuscript, focusing on the most solid result and incorporating either functional data or orthogonal structural. best regards, Tatiana Kutateladze

Reviewer #2

(Remarks to the Author)

In the manuscript entitled "Mechanistic insights into histone recognition and H3K14 acetylation by the NuA3 histone acetyltransferase complex", Shi et al. present the cryo-EM structures of the yeast NuA3 complex in three different states: apo, bound to acetyl-CoA, and bound to both acetyl-CoA and an H3 tail peptide. Compared to previous studies that only resolved the catalytic core, these new structures provide more information about the yeast NuA3 complex, which is exciting. However, the structure that includes both the H3 tail and acetyl-CoA, the most critical for understanding substrate recognition, was unfortunately obtained at the lowest resolution, limiting the authors' ability to convincingly interpret the molecular mechanism.

Major Comments:

1. In Fig. 4, the resolution of the H3 tail needs to be significantly improved to confidently determine the register. Without clearer density, any discussion of the molecular recognition mechanism remains speculative and potentially misleading.
2. The authors were unable to resolve the density corresponding to the region around H3K4me3, which is a key determinant of NuA3 specificity. This limits mechanistic interpretation regarding histone tail recognition.
3. The use of a synthetic H3K14M peptide to mimic a pre-catalytic state may not fully recapitulate the native interaction. The authors are encouraged to consider capturing the ternary complex using a catalytically dead mutant with a wild-type H3K4me3 peptide to improve map quality and biological relevance.
4. In Fig. 6a, there is a serine residue preceding H3K23, which is similar in size to glycine. The authors should address this point when emphasizing the importance of Gly13 in determining substrate specificity.

Minor Comments:

1. Many references are mis-cited or vague, e.g. in lines 56–58 and line 71. The authors should go through all the citations carefully.
2. Line 98: The ET domain is discussed but not clearly labeled in the figure. It is better to label all domains in the structure to improve clarity and accessibility for the reader.
3. The color scheme in Fig. 3 is difficult to distinguish. The authors should consider using more contrasting or clearly distinguishable colors.
4. Line 190. acetyl-CoA is missing.

Reviewer #3

(Remarks to the Author)

Version 1:

Reviewer comments:

Reviewer #1

(Remarks to the Author)

The authors have adequately addressed all my comments.

Reviewer #2

(Remarks to the Author)

The authors have performed additional assays and collected further data, substantially improving the resolution and enabling confident model building. I have no further questions and consider the work complete and ready for publication.

Reviewer #3

(Remarks to the Author)

Reviewer #1 (Remarks to the Author):

In this work, Shi and colleagues report the cryo-EM structure of the yeast NuA3 complex in the apo state and in the presence of acetyl-CoA and histone H3K4me3 peptide (aa 1-21 of H3). Three subunits, Sas3, Nto1 and Eaf6 are fully resolved in the structure, two subunits, Taf14 and Yng1 are partially resolved with the PHD finger and the YEATS domain missing, and no density is observed for the sixth subunit, Pdp3. Despite the missing elements, the structure provides very helpful information regarding some aspects of the complex assembly and how the region of H3 primes at the Sas3 active site for acetylation of H3K14. The major weakness of this work is the incomplete complex structure which may need revision in future. The data on the H3 peptide-acetyl-CoA-Sas3 contact(s) is a solid and important finding here, but it's a relatively small set of results, and I am afraid a considerable number of biological/functional experiments is needed to corroborate the contacts and complete this work. Alternatively, if functional approaches are not feasible, additional structures should be included, which might be more straightforward to this group as the NuA3 complex is already available.

The structural data are of excellent quality, some conclusions are justified though some are unconvincing, still I support this work and believe the manuscript can be effectively reworked and with inclusion of additional data will be a valuable contribution to the epigenetics field.

Our response: We thank the reviewer for the positive evaluation and constructive suggestions on our manuscript. We have adequately addressed all the comments in our revised manuscript and revised figures. As suggested by the reviewer, we performed functional assays to confirm our structural observations.

To validate our structural observations and *in vitro* enzymatic assays, we introduced two single mutations, E452Q and L469R, which have been shown to abolish the catalytic activity of NuA3 in our assays, into the SAS3 gene. We then assessed their effects on basal acetylation levels in *S. cerevisiae* using Western blot analysis. Consistent with our *in vitro* results, these mutations led to a decrease in acetylation

levels compared to wild-type yeast strains (Fig. 5d).

A few specific comments:

1. Abstract: please remove the phrase “In sharp contrast with previously... complexes”. The negative view on others’ findings never perceived well, especially if you are somewhat incorrect. And overall, for the entire text, I would suggest writing it in a way to convey how your discovery [taking advantage of the previous knowledge] moves the epi field forward.

Our response: The phrase ‘in sharp contrast with ...’ has been removed in our revised manuscript.

2. First sentence of Intro is incorrect and indicates a limited knowledge of the chromatin field.

Our response: Thanks for pointing this out. The sentence has been revised to ‘Chromatin organizes the eukaryotic genome by compacting DNA and regulating its accessibility. Its fundamental unit, the nucleosome, consists of ~147 base pairs of DNA wrapped around a histone octamer’.

3. Line 65, mentioning H3K4me3 and H3K36me3- who binds these marks and why these interactions are important for the function of the complex? ref 23-25 are great but quite old papers. Numerous more recent studies have to be discussed here, including those that show that the PHD finger of Yng1 is essential for the NuA3 function(s). The PHD finger is not observed in the current structure, even in the presence of the H3K4me3 peptide, however this cannot be the reason for not discussing Yng1.

Our response: As suggested, we have expanded the manuscript to include a more detailed description of the functions of various domains, including the PHD finger of Yng1 (Lines 59–71). Relevant references have been updated and additional recent studies have been cited accordingly.

4. Similarly to #3, no mentioning of the YEATS domain of Taf14 and the importance of its binding to H3K9acetyl for the NuA3 complex.

Our response: As suggested, we have added a discussion of the YEATS domain of Taf14 and its role in recognizing H3K9acetyl modifications (lines 66-69). The references

have been revised and updated accordingly.

5. Overall, Intro needs to be substantially expanded with an expert level discussion of up-to-date studies.

Our response: Thanks for pointing this out. We have now substantially expanded the introduction and incorporated recent literature to provide a more comprehensive and up-to-date discussion.

6. Lines 94, 98, the word 'extensively' is not needed

Our response: The word 'extensively' has been removed from lines 94 and 98

7. Fig. 1b- please add labels

Our response: Labels have been added to Fig. 1b.

8. Fig.1, please add schematics of all six subunits of the NuA3 complex and indicate which parts are observed in the structure.

Our response: Schematics of all six NuA3 subunits have been included in Fig. 1 (Fig. 1g), with clear indication of the regions solved and unsolved in the structure. The figure legend has been updated accordingly.

9. Line 139, please add refs to PZP domain, and because PZP is a well-studied domain, the paragraph lines 138-149 could be condensed into a couple of sentences.

Our response: We appreciate the reviewer's helpful suggestions. We have added appropriate references to indicate that the PZP domain has been extensively studied. The paragraph in question (lines 138–149 in previous version) primarily focuses on the Nto1 PZP domain and its interactions with other NuA3 subunits. We believe these descriptions are important for maintaining clarity and accessibility for a broader readership, and therefore have chosen to retain this level of detail.

10. Lines 150-161. This paragraph needs to be reworked and shortened to essentially one sentence. Ref 46 discovered that the hxhxx motif (referred to EBM, ET binding motif) is present in both Yng1 and Sas3, and that the ET domain of Taf14 binds EBM-containing peptides derived from both Yng1 and Sas3. Both interactions are thoroughly investigated, and the authors of ref 46 put forward the idea that either Yng1 or Sas3 can be the intra-complex ligands of Taf14. Ref 46 reports the structure of the ET domain with only one of those peptides, Yng1 peptide. The way how the paragraph on lines 150-161 is written is misleading: omitting Sas3 and focusing on Yng1. Conversely, it should be written 'In support of previous studies (ref 46, please also add Nguyen 2023, BBA, Klein 2022, Nat Comm, and Chen 2020, Nat Comm) the ET domain of Taf14 binds to a well-recognized ... motif, present in Sas3' or something similar.

It will be appropriate in discussion to come back to ref 46 stating that '... from our structure, indeed Sas3 interacts with Taf14 but Yng1 is far away...' or something similar. Also appropriate (and valuable) to discuss possibilities of conformational changes in the complex that would allow Yng1 to interact with Taf14. [why this motif is present in

Yng1?] [of note, the same EBM motif is also present in Nto1, unpublished]

Our response: We thank the reviewer for pointing out this mistake. The paragraph (lines 150–161 in the previous version) has been revised to: “In support of previous studies, the ET domain of Taf14 binds to a well-characterized EBM motif present in Sas3.” To keep consistency, the corresponding panels (Fig. 3g, 3h) and their figure legends have been removed accordingly. As suggested, we have also added a sentence to the Discussion (lines 273-276): “In our structure, Sas3 interacts with Taf14, whereas the neighboring residues of the EBM motif in Yng1 are located far from the ET domain. It is possible that conformational changes under certain conditions might allow Yng1 to interact with Taf14, and further studies are needed to test this possibility.”

11. Line 191, no rationale is given to use H3K4me3 (1-21) peptide. If the rationale was to aid in detecting PHD of Yng1, it needs to be stated. Overall, the use of this peptide is the major weakness of this work. Why not H3K4me3K27acK36me3 or H3K9ac? Or unmodified H3 (to engage with PZP of Nto1)

Our response: Thank you for this valuable comment. As you know, Taf14 is a shared subunit of several complexes, including TFIIIF, TFIID, RSC, INO80, SWI/SNF, as well as NuA3. Previous work, including from the reviewer’s group, has demonstrated that Taf14 can recognize H3K9ac peptides, substantially advancing our understanding of how Taf14 interacts with acetylated histone tails. However, the precise role of H3K9ac within each protein complex, particularly in NuA3 complex, remains insufficiently characterized, and a detailed functional understanding is still lacking. In this study, our focus was on defining the recognition mechanism of H3 by NuA3 in a clear and unambiguous manner. For this reason, we chose the H3K4me3 (1–21) peptide, which is the canonical substrate recognized by the PHD finger of Yng1 and thus provides a reliable probe for structural and biochemical analysis.

Nonetheless, we have now included a few sentences (lines 162-170) in the revised manuscript to explain why the H3K4me3 (1-21) peptide was used in this study.

12. Related to #11, the use of any of these peptides may not provide a true mechanism of the complex function. The nucleosome must be used here, and better PTM containing. In the current structure H3K4me3peptide-acetyl-CoA- bound complex, the only solid unambiguous result is the contacts within the H3(9-18)-Sas3 active site; the rest of the mechanistic data may be completely off in the NCP-complex structure.

Our response: We thank the reviewer for this insightful suggestion. In response, we explored two strategies to assemble the NCP–NuA3 complex for structural studies: (1) wild-type NuA3 with a 147-bp NCP carrying an H3K14M mutation, and (2) NuA3 (E452Q) with wild-type 167-bp NCP in the presence of acetyl-CoA. We collected two datasets using a Titan Krios and processed the data accordingly. Unfortunately, neither approach was successful. The interaction between NCP and NuA3 appears to be

highly dynamic, preventing us from resolving their interaction details despite applying various data-processing strategies, including focused classification. Please see below:

We fully agree that a complete nucleosome-bound structure would provide the most definitive mechanistic insights; however, obtaining such a structure for this complex is not currently feasible. Importantly, numerous studies—including several pioneering works from the reviewer's own lab—have successfully employed histone peptides as surrogates for nucleosomes. While nucleosomes do provide additional contacts with histone-modifying enzymes, to date no evidence has demonstrated a difference in recognition between histone peptides and the corresponding region in nucleosomes by these enzymes.

13. Line 222, re PZP of Nto1- would be good to superimpose with other PZP domains

Our response: Thank you for this valuable suggestion. As mentioned in response to #9, we have performed structural superposition (Fig. 3b) of the Nto1 PZP domain with other PZP domains, including those of BRPF1, AF10, PHF14, and JADE1.

14. Line 272, please discuss YEATS domain of Taf14 and its role

Our response: As recommended, the role of the YEATS domain of Taf14 has been discussed in the revised manuscript (lines 231-232), with appropriate citations.

15. Lines 275-287, the G13R mutation data to explain NuA3 selectivity are unconvincing and naïve. The RKac motif may play a role for the YEATS domain.

Our response: Thanks for this insightful comment. As we mentioned in the abstract, the specificity is not determined by a single amino acid in the histone H3 peptide. The substrate specificity is determined by several factors, including the N-terminal hydrophobic interactions, the interaction between the H3K4me (aa 1-4) and the PHD1 subdomain of Yng1, although unsolved in our structure, and the hydrogen bonds formed between the H3 and the NuA3 complex. A few sentences have been added to the revised manuscript to make a better interpretation (lines 244-250).

16. Line 305, this phrase is incorrect- PZP domain binds exclusively to unmodified H3; methylation of K4 inhibits this interaction. [I hope that this is a misspelling error rather than the rationale to use H3K4me3 peptide in this work]

Our response: We thank the reviewer for pointing this out. We agree that these statements were incorrect and have removed them, along with Figure S9, from the revised manuscript. These statements were not essential to our study, and their deletion does not affect any of the conclusions presented.

17. Lines 309-310, this phrase is again incorrect, and it's unlikely that the PZP of Nto1 lost its ability to bind unmodified H3.

Our response: We thank the reviewer for pointing this out. As there is currently no direct experimental evidence confirming whether the *S. cerevisiae* Nto1-PZP can or cannot bind unmodified H3, we have removed the corresponding statements and the corresponding Extended Figure 9b from the revised version to avoid overinterpretation.

18. Overall, I would suggest reworking the manuscript, focusing on the most solid result and incorporating either functional data or orthogonal structural.

best regards, Tatiana Kutateladze

Our response: We appreciate this constructive advice. We have reworked the manuscript to focus on the most robust structural findings, and we acknowledge the need for complementary functional and/or structural approaches to further extend these insights.

Reviewer #2 (Remarks to the Author):

In the manuscript entitled "Mechanistic insights into histone recognition and H3K14 acetylation by the NuA3 histone acetyltransferase complex", Shi et al. present the cryo-EM structures of the yeast NuA3 complex in three different states: apo, bound to acetyl-CoA, and bound to both acetyl-CoA and an H3 tail peptide. Compared to previous studies that only resolved the catalytic core, these new structures provide more information about the yeast NuA3 complex, which is exciting. However, the structure that includes both the H3 tail and acetyl-CoA, the most critical for understanding substrate recognition, was unfortunately obtained at the lowest resolution, limiting the authors' ability to convincingly interpret the molecular mechanism.

Our response: We sincerely thank the reviewer for the positive evaluation and constructive suggestions, which have greatly helped us improve the manuscript. Over the past several weeks, we have focused on enhancing the map quality, and the resolution of the NuA3–H3–acetyl-CoA complex has been improved from 3.8 Å to 3.2 Å, making our interpretation more convincing.

Major Comments:

1. In Fig. 4, the resolution of the H3 tail needs to be significantly improved to confidently determine the register. Without clearer density, any discussion of the molecular recognition mechanism remains speculative and potentially misleading.

Our response: We thank the reviewer for these insightful comments. As suggested, over the past few weeks, we optimized grid preparation and data collection, and collected a new cryo-EM dataset on Titan Krios. Data processing yielded a 3.2 Å resolution for the NuA3–acetyl-CoA–H3 complex (the previous resolution is 3.8 Å). The H3 tail density is now much clearer, allowing us to refine the register with confidence (Fig. 4e). The new higher-resolution cryo-EM map further supports the recognition mechanism in our previous version. We have updated all relevant figures (Fig. 4e and Extended Data Fig. 7), including the cryo-EM workflow (Extended Data Fig. 6), FSC curve, local resolution map, and density–model fit (Extended Data Fig. 7).

Based on our higher-resolution structure, we have made slight revisions to our descriptions of the interactions between H3 and NuA3, as well as between acetyl-CoA and NuA3 (lines 187-189).

2. The authors were unable to resolve the density corresponding to the region around H3K4me3, which is a key determinant of NuA3 specificity. This limits mechanistic interpretation regarding histone tail recognition.

Our response: We agree with the reviewer that the density around H3K4me3 could not be resolved, most likely due to the flexibility of the Yng1-PHD finger. The recognition mechanism of H3K4me3 by Yng1-PHD has already been well established by the seminal work of Prof. Allis C.D.'s lab (*Mol Cell*, 2006). While we could not visualize this interaction in our structure, these prior studies provide a clear framework for understanding H3K4me3 (residues 1-4) recognition, and thus we believe the lack of direct density of the region around H3K4me3 in our map does not preclude mechanistic interpretation.

3. The use of a synthetic H3K14M peptide to mimic a pre-catalytic state may not fully recapitulate the native interaction. The authors are encouraged to consider capturing the ternary complex using a catalytically dead mutant with a wild-type H3K4me3 peptide to improve map quality and biological relevance.

Our response: Following the reviewer's suggestion, we reconstituted a NuA3 (E452Q)–acetyl-CoA–H3 (wild-type) complex and collected a new cryo-EM dataset on Titan Krios. After data processing, we found that in this complex, no electron density corresponding to the histone tail was observed around the H3-binding cleft (highlighted by a white dashed box in the figure below). Although the underlying reason is not fully clear, we believe that the strategy described in our manuscript provides a more effective means of capturing the pre-reaction state of the complex.

4. In Fig. 6a, there is a serine residue preceding H3K23, which is similar in size to glycine. The authors should address this point when emphasizing the importance of Gly13 in determining substrate specificity.

Our response: We thank the reviewer for this insightful comment. We agree that substrate specificity cannot be attributed to a single residue. In the revised text, we clarify that specificity is determined by multiple factors, including N-terminal hydrophobic contacts, interactions—although not resolved in our structure—between H3K4me (residues 1–4) and the Yng1 PHD finger, and the hydrogen bonding between H3 and NuA3 (lines 244-250).

Minor Comments:

1. Many references are mis-cited or vague, e.g. in lines 56–58 and line 71. The authors should go through all the citations carefully.

Our response: Thanks for pointing this out. We carefully checked all references and corrected mis-citations and vague attributions. For clarity, we have revised line 71 to “... as well as on the SAGA complex⁴⁸⁻⁵⁰, the yeast NuA4 complex⁵¹⁻⁵⁵, and human TIP60 complex⁵⁶⁻⁵⁸”

2. Line 98: The ET domain is discussed but not clearly labeled in the figure. It is better to label all domains in the structure to improve clarity and accessibility for the reader.

Our response: As suggested, we have labeled all domains, including the ET domain, in the revised figures for clarity.

3. The color scheme in Fig. 3 is difficult to distinguish. The authors should consider using more contrasting or clearly distinguishable colors.

Our response: We thank the reviewer for this helpful suggestion. We have changed the color of the EPL1-like subdomain from light green to cyan. In addition, to enhance

clarity, the Nto1 subunit in Figure 3a is now displayed in cylindrical helical representation. For Figure 3c, the colors are clearly distinguishable in the original images; however, the resolution may have been automatically reduced during manuscript submission, making the differences less apparent. Therefore, we did not adjust the coloring in Figure 3c.

4. Line190. acetyl-CoA is missing.

Our response: acetyl-CoA has been added (line 172).

Reviewer #3 (Remarks to the Author):

Our response: We sincerely thank Reviewer #3 for the positive evaluation and constructive suggestions on our manuscript.

REVIEWERS' COMMENTS

Reviewer #1 (Remarks to the Author):

The authors have adequately addressed all my comments.

Our response: Thanks.

Reviewer #2 (Remarks to the Author):

The authors have performed additional assays and collected further data, substantially improving the resolution and enabling confident model building. I have no further questions and consider the work complete and ready for publication.

Our response: Thanks.

Reviewer #3 (Remarks to the Author):

Our response: Thanks.